# Use of Oleogels to Replace Margarine in Steamed and Baked Buns

**DOI:** 10.3390/foods10081781

**Published:** 2021-07-31

**Authors:** Santiago Bascuas, Pere Morell, Amparo Quiles, Ana Salvador, Isabel Hernando

**Affiliations:** 1Department of Food Technology, Universitat Politècnica de València, 46021 Valencia, Spain; sanbasve@upvnet.upv.es (S.B.); pemoes@upv.es (P.M.); mquichu@tal.upv.es (A.Q.); 2Institute of Agrochemistry and Food Technology (IATA-CSIC), 46980 Valencia, Spain; asalvador@iata.csic.es

**Keywords:** hydrocolloids, bakery products, oil structuring

## Abstract

Bakery products are usually formulated with solid fats, like margarines and shortenings, which contain high levels of saturated and trans-fatty acids and have negative effects on human health. In this study, hydroxypropyl methylcellulose (HPMC) and xanthan gum (XG) were used as oleogelators to prepare oleogels, using sunflower and olive oil, as substitutes for margarine in baked or steamed buns. The effect of oleogels on the physical properties of the buns was evaluated by analyzing the crumb structure, specific volume, height, and texture. In addition, a triangular discriminatory sensory test was conducted, and lipid digestibility was assessed through in vitro digestion studies. Replacement of margarine with oleogels produced steamed buns with no differences in the crumb structure, volume, height, and texture; however, in baked buns, a less porous and harder structure was produced. No differences in texture were observed between the margarine buns and buns made with oleogels when the triangular test was conducted. The extent of lipolysis was not affected when margarine was replaced by oleogels in the baked and steamed buns. The results suggest that using oleogels instead of margarine in buns could represent an interesting strategy to prepare healthier bakery products.

## 1. Introduction

Bread has been an important staple food in many aspects of humanity and civilization over the centuries. Commonly, dough as an aqueous colloidal dispersion comprises both hydrophobic (fats and shortenings) and hydrophilic (flour and sugar) components. During bread making, the components of dough form a three-dimensional structure, which traps air bubbles throughout baking [1].

The bread industry has made many efforts to create new products with different appearance, shape, flavor, and nutritional value to satisfy consumer demands (Martínez-Monzó et al., 2013). Sweet breads, such as buns, are an interesting choice established in the baking industry. Many food ingredients and bun-making processes are used to formulate a diversity of bun bread [2,3,4]. For example, incorporating dairy products is associated with a variety of nutritional value and functional properties, such as texture, sensorial properties, and storage characteristics [5,6,7]. However, many bakery goods are formulated with solid fats, like margarines and shortenings, which contain high levels of saturated and trans-fatty acids [8,9]. The increased saturated fat and trans-fat intake has negative effects on chronic diseases, such as obesity, cardiovascular disease, and diabetes [10,11].

The replacement of margarines and shortenings in bakery food products is difficult to achieve without negatively affecting physical properties, because they play key roles tenderizing the crumb, retaining long-term softness, keeping quality, and extending the shelf life [12,13]. However, bakery products seem like suitable candidates for saturated and trans-fat replacements or reduction, because of their high frequency of consumption, and these changes would confer substantial health benefits [14,15].

Oleogels are an important technological advance in food science because of their versatility, easy processing, and affordability [16]. Oleogels are defined as gel systems where a continuous liquid oil phase is immobilized in a network of self-assembled molecules of an oleogelator or a combination of gelators. They can be a viable alternative to replace solid fats in buns, providing them with a healthier nutritional profile. There are only a few studies on hydrocolloids-based oleogels as a shortening alternative in bakery products, such as pea protein–xanthan gum (XG) foam-based oleogels in cakes (Mohanan et al., 2020), foam-structured HPMC oleogels in muffins [17], and MC-XG oleogels in cakes [18]. Recently, the microstructure, rheological behavior, physical, and oxidative stability of oleogels prepared using HPMC and XG were characterized, resulting in oleogels with enhanced rheological properties and thermostability with low oil losses (˂10%) after 35 days of storage [19,20]. These oleogels made with HPMC and XG can be a promising alternative to using solid fat in bakery products. Moreover, studies linking the role of oleogels made with HPMC and XG as structuring agents in the functionality and digestibility of buns have not been found.

The focus of this study is to compare the behavior and properties of buns elaborated with olive or sunflower oil oleogels, made with HPMC and XG as oleogelators, to replace margarine in steamed and baked buns. The results from the analysis of crumb structure, specific volume, and textural properties will provide an understanding of the structural aspects that underlie the incorporation of oleogels in bakery products. Sensory and in vitro digestibility studies will also inform the use of oleogels for food formulation, both at industry and homemade levels.

## 2. Materials and Methods

### 2.1. Ingredients

Hydroxypropylmethylcellulose (HPMC ‘K4M’; 4000 cP) was provided by Dow Chemical Company (Midland, MI, USA) and xanthan gum (XG; Satiaxine CX 931) by Cargill R&D (Vilvoorde, Belgium). Water (Bezoya, Segovia, Spain, with a calcium content 6.32 mg/L), olive oil (fatty acids composition: SFA:12, MUFA: 71, PUFA: 8, Coosur, Sevilla, Spain), high oleic sunflower oil (fatty acids composition: SFA: 10, MUFA: 65, PUFA: 25, Carrefour, Madrid, Spain), wheat flour (Comercial Gallo S.A.U., Córdoba, Spain), fresh yeast (Lesaffre Ibérica, Valladolid, Spain), salt (Consum, Valencia, Spain), sugar (Azucarera, Madrid, Spain), and whole milk (3.6 g/100 mL fat) (Consum, Valencia, Spain) were purchased in supermarkets. Margarine (SFA: 56–66) was supplied by Gracomsa (Catarroja, Valencia, Spain).

### 2.2. Oleogel Preparation

Oleogels (olive and sunflower oil) were prepared following the procedure used by Bascuas et al. [20]. HPMC (2 g) was dispersed in 76.8 g cold water and mixed using a food processor (TM31 Thermomix, Vorwrek, Wuppertal, Germany). Subsequently, 1.2 g of XG was added to the resulting aqueous solution and mixed for 5 min at 300 rpm. Then, 120 g of oil was gradually added and homogenized in the processor for 5 min at 300 rpm. The emulsions were spread on a Teflon tray (42 cm × 35 cm × 0.13 mm, Pritogo, Salgen, Germany) using a silicone pastry bag (Kurtzy, Coimbatore, India) and plastic nozzle (diameter of 4.5 cm, TESCOMA, Carrefour, Madrid, Spain). The emulsions were dried using forced convention air in an oven (KB115, BINDER, Tuttlingen, Germany) at 80 °C for 3 h 30 min. This was the minimum time needed to reach constant dry weight (moisture: 1.75 ± 0.51%) under the indicated conditions. The dried products were ground in a grinder (Moulinex A320R1, Paris, France) for 4 s to produce the oleogels.

### 2.3. Bun Making

Six samples were formulated containing 53.5 g wheat flour, 26 g milk, 10 g fat (margarine or oleogel), 6 g sugar, 3.5 g fresh yeast, and 1 g salt. A food processor (TM31 Thermomix, Vorwrek, Wuppertal, Germany) was used to mix the ingredients. First, the sugar and milk were mixed at 37 °C for 2 min at 200 rpm. After, the fresh yeast was added and mixed in the processor for 5 s at 300 rpm. The wheat flour and salt were added and mixed for 15 s at 1100 rpm. Finally, the margarine, olive oil oleogel, or sunflower oil oleogel was added, and the kneading function was used at 500 rpm for 3 min to obtain dough C (margarine), dough O (olive oleogel), and dough S (sunflower oleogel), respectively. Then, 150 g of dough was spread on aluminum trays (0.5 L, Alibérico Food Packaging, Madrid, Spain) and the dough was left to rise for 1 h at 28 °C. The doughs were cooked using two conditions: baked (B) in an oven (Electrolux, model EOC3430DOX, Stockholm, Sweden) at 180 °C for 30 min, or steamed (S) in a Thermomix at 90 °C for 30 min using the Varoma (vapor) function. Six buns (BC, SC, BO, SO, BS, and SS; first letter indicating type of cooking and second letter indicating use of margarine or oleogel) were kept covered at room temperature for 1 h and then analyzed.

### 2.4. Analysis of the Crumb Structure

The samples were cut into vertical slices of 15 mm thickness and scanned (with a resolution of 300 dpi) using a computer scanner Epson Perfection 1250 (Epson America Inc., Long Beach, CA, USA). The crumb cellular structure was analyzed using the software ImageJ (National Institutes of Health, Bethesda, MA, USA). The image was cropped to a 2 × 4 cm section, on which the analysis was performed. The image was split into color channels and the contrast was enhanced; the image was then binarized after a gray-scale threshold. The parameters calculated were air cell density (number of cells per field), air cell area (mm^2^), cell circularity, and total air cell area within the crumb (%). Measurements were performed in triplicate.

### 2.5. Specific Volume and Height

The specific volume of bread loaves was measured at room temperature using the rapeseed seed displacement method (AACC International Method 10–05.01) [21]. The maximum bread height was measured from the cross section of the image scanned using the software ImageJ.

### 2.6. Texture Measurements

A TA-TX plus Texture Analyzer (Stable Micro Systems, Ltd., Godalming, UK) with the Texture Exponent Lite 32 software (version 6.1.4.0, Stable Micro Systems) was used to determine the texture properties of the samples. Measurements were performed in triplicate on eight cube samples (15 × 15 × 15 mm) taken from the central crumb of each bread.

Texture profile analysis (TPA) was performed with a 35 mm diameter aluminum plate (P/35) using a test speed of 1 mm/s with a strain of 40% of the original cube height and a 5 s interval between the two compression cycles. The parameters obtained from the curves were hardness, springiness, cohesiveness, and chewiness.

### 2.7. Sensory Analysis

A triangular discriminatory test was performed with a panel of 20 panelists to determine whether there were significant differences between two samples of buns. One triad was prepared per taster, who had a duplicate and a different sample. Each sample was coded with three random digits and was presented an equal number of times in each of the possible positions: BAA, AAB, ABA, ABB, BBA, and BAB in random order following a Williams design. Each panelist evaluated the samples of the triad and marked the sample they considered different. Baked and steamed buns were analyzed separately, and the buns made with the different oleogels were presented against their respective control buns. Results were analyzed following a unilateral hypothesis with a significance level of 5% [22].

### 2.8. In Vitro Digestion

In vitro digestion of buns was performed according to the procedures described by Diez-Sánchez et al. [23] with some modifications. Solutions of simulated salivary fluid (SSF), simulated gastric fluid (SGF), and simulated intestinal fluid (SIF) were elaborated according to the compositions described in the INFOGEST 2.0 protocol [24]. To mimic human physiological conditions, the analysis was conducted with a controlled temperature (37 °C) and agitation (150 rpm).

First, 7.5 g of each bun sample was ground using a hand blender (Ufesa, model BP4566, Barcelona, Spain) and then 6 mL of SSF + α-amylase (Sigma A3176), 27.5 μL of CaCl_2_, and 1.472 mL of distilled water were added and mixed by hand for 2 min to simulate mastication. Second, in the gastric stage, 24 mL of SGF + pepsin (Sigma P7000) and 12 μL of CaCl_2_ were added. The pH was adjusted to 3 using 1 M HCl, and the volume of distilled water necessary for a total volume of 30 mL was added. The mixture was incubated at 37 °C for 1 h under agitation. Third, for the intestinal stage, 12 mL of SIF + pancreatin (Sigma P1750, 4xUSP), 67.5 μL of CaCl_2_, 12 mL of SIF + bile salts (Sigma B8631) [25], and 12 mL of SIF + lipase (Sigma L3126; 2000 U/mL) were added. The pH was adjusted to 7 using 1 M NaOH. The mixture was incubated at 37 °C for 2 h under agitation. The measurement was conducted in triplicate.

### 2.9. Free Fatty Acid Release

The release of free fatty acids (FFA) of the samples was recorded during in vitro intestinal digestion using a pH-stat automatic titration unit (Mettler-Toledo DL 50, Greinfensee, Switzerland). This method is designed to simulate lipid digestion within the small intestine (where most lipid digestion normally occurs) and is based on measurements of the FFA released from lipids (usually triacylglycerols) after lipase addition [26]. The pH was automatically monitored and maintained at pH 7.0 by titrating appropriate amounts (mL) of the NaOH solution (0.25 M). The volume of NaOH added to the sample was recorded and used to calculate the concentration of FFA by lipolysis using Equation (1) [27]:(1)$$\mathrm{FFA}\;\left(\%\right)=100\times \frac{V_{\mathrm{NaOH}}\times {\;m}_{\mathrm{NaOH}}\times {\;M}_{\mathrm{lipid}}}{W_{\mathrm{lipid}}\times 2}$$
where V_NaOH_ is the volume of NaOH (L) added during the digestion process to neutralize the FFAs generated, m_NaOH_ is the molarity of the NaOH titrant, M_lipid_ is the average molecular weight of solid fat (200 g/mol for margarine and 282 g/mol for olive and sunflower oil oleogels), and W_lipid_ is the total weight of oil in the digestion system.

The experimental data of FFA released were fitted with the empirical model following Equation (2) [27]:(2)$$\mathrm{FFA}\;\left(\%\right)=[{\left(\mathrm{FFA}\right)}_{\max}\times \;t]/\left(B+t\right)$$
where % FFA and % FFA_max_ are the % FFA released at the time t and at the “pseudo-equilibrium”, respectively, and B is the minimum time needed to reach half the lipolysis half time, that is % FFA_max_/2.

The initial rate (K) of FFA release can be calculated using model (3):(3)$$K={\left(\mathrm{FFA}\right)}_{\max}\;/B$$

### 2.10. Statistical Analysis

Results were statistically analyzed using the analysis of variance (ANOVA) to study the effects of fat. The least significant differences (LSD) were calculated at a level of significance *p <* 0.05 using the statistical program Statgraphics Centurion XVI.II (StatPoint Technologies, Inc., Warrenton, VA, USA).

## 3. Results and Discussion

### 3.1. Analysis of the Crumb Structure

The appearance of the bun slides is shown in Figure 1. The baked buns (Figure 1: BC, BO, and BS) had a hard and dark crust, whereas the steamed buns (Figure 1: SC, SO, and SS) presented a thin white crust; both had a white and bright crumb. Comparing BC (Figure 1: BC) with BO and BS (Figure 1: BO and BS), BC seemed to present a more aerated and open structure, whereas BO and BS showed a more compact and denser crumb structure, with a smaller number of cavities. However, these differences in the appearance could not be observed between the steamed samples (Figure 1: SC, SO, and SS).

Table 1 and Table 2 show the crumb cell structure parameters of baked and steamed buns, respectively. BC had significantly higher values (*p <* 0.05) in the total cell area than BO and BS. Moreover, BC showed higher significant (*p <* 0.05) values than BS samples in cell circularity. No significant (*p >* 0.05) differences could be observed in the other cell structure characteristics (cell density and cell area). Regarding steamed buns (Table 2), no significant (*p >* 0.05) differences could be observed in the cell structure characteristics (cell density, cell area, cell circularity, and total cell area) between BC, BO, and BS.

Solid fat plays a key role in the stabilization of air bubbles, forming a film in the air–matrix interface [28]. The formation of a structured network provided by oleogels appears very effective in air cell incorporation and stabilization in buns. This can explain the similar values of cell density and cell area observed when buns are formulated with oleogel or margarine. This demonstrates the ability of oleogels, formulated with HPMC and XG, to impart a similar structure like the control bun made with a saturated fat, while improving the lipid profile of the fat.

### 3.2. Specific Volume

Specific volume is one of the most critical parameters to assess aerated baked goods’ quality that strongly affects consumer acceptance [29]. Lipids have an important impact on dough volume [30]; hence, volume loss is a technological difficulty when replacing conventional fat in the baking industry with fat replacers.

Table 1 shows the effect of margarine and oleogels on the specific volume and height of baked buns. No significant differences (*p >* 0.05) were found in the specific volume and height between BC, BO, and BS. Using the hydrocolloids in the oleogels preparation would favor the formation of a gel network structure during baking, improving the specific volume by expanding the gas cells without them collapsing, as explained by other studies [31,32]. Table 2 displays the specific volume and height of steamed buns. SO and SS showed no significant (*p >* 0.05) differences with SC. Other authors [33] also reported no significant differences in the specific volume of sweet bread replacing 75% of butter with oleogel prepared with candelilla wax.

Thus, the solid-like structure of the oleogel used in this study appears a viable alternative to resemble the functionality of conventional solid fats to maintain the quality standards of buns in terms of volume and height.

### 3.3. Texture

The textural properties of BC, BO, and BS are summarized in Table 1. BO and BS exhibited significantly higher hardness values (*p <* 0.05) than BC. As expected, these results are in line with the crumb structure results already analyzed. BC presented the highest total cell area values, which implies a more aerated structure that offers less resistance to compression. This relationship has been observed in cakes prepared with other fat replacers [34,35]. Moreover, Oh et al. [17] also reported higher hardness values in muffins elaborated with 100% replacement of shortening with HPMC oleogels when compared to control muffins. The authors attributed these differences to the lower values in the total porosity of the samples made with oleogels. Chewiness values presented a similar trend as hardness values; BO and BS had values higher than BC. Martínez-Cervera et al. [36] reported an increase in chewiness values when muffins were made using cellulose emulsions instead of margarine. Regarding springiness, BC presented higher significant (*p <* 0.05) values than BS. The cohesiveness values showed no significant (*p >* 0.05) differences between BC, BO, and BS. These results agree with the specific volume analysis, where baked buns did not present significant (*p >* 0.05) differences.

As presented in Table 2, SO and SS exhibited no significant (*p >* 0.05) differences in texture properties compared to SC. These results are consistent with the appearance of in steamed bun slides, where the SO and SS buns exhibited a similar shape and aerated crumb structure, whereas the crumb cell structure showed no differences between the buns.

Texture parameters play a key role in the quality of leavened buns where the geometric and mechanical properties heavily depend on its cellular structure [37]. Buns formulated using hydrocolloids-based oleogels lead to a well-developed porous crumb structure and could represent a suitable strategy to imitate the functionality of margarine in buns.

### 3.4. Sensory Analysis

To know if there were differences between buns prepared with or without oleogels, a sensory discriminative test (triangle test) was performed; the triangle test is one of the most widely used analysis when products evaluated are sufficiently homogeneous [38]. The descriptors or attributes used in the triangle test in this study were the appearance of the crumb, texture, and taste. According to ISO 4120:2004, 20 total triangle test responses require 11 correct responses to be significant at the 95% confidence level [22]. The results indicated panelists could differentiate between buns made with margarine and buns made with oleogels, both baked and steamed.

Specifically, when baked buns were made using olive or high oleic sunflower oleogels, the panelists detected significant differences (*p <* 0.05) in the appearance of crumb and taste attributes when compared with the control buns. Furthermore, no significant (*p >* 0.05) differences were observed in texture by panelists. For steamed buns, a similar tendency was observed.

Texture is an essential sensory attribute that can determine the product quality and play a vital role regarding the acceptability of breads among consumers. In this study, the sensory results show full replacement of margarine using hydrocolloids-based oleogels did not affect the texture; this would be an important advantage of the oleogels, which could be used to obtain buns with high sensory quality in terms of texture.

### 3.5. Free Fatty Acid Release

All samples mostly showed a fast release of FFA in the first 10 min of the intestinal digestion (Figure 2), followed by a more gradual increase after the 10 min point, reaching a relatively constant final value. Because fat is insoluble in water, lipase has to adsorb to the oil−water interface for the enzymatic reaction to occur. Therefore, fat digestion is controlled by the ability of lipase to bind to emulsion interfaces. The saturation on the extent of lipolysis may be because the FFA and intermediate products have high surface activity and adsorb to the surface of oil droplets. At a sufficiently high concentration, they displace the lipase molecules from the oil-water interface, thus inhibiting the lipase activity [39].

Table 3 and Table 4 describe the kinetic parameters of FFA released during the in vitro digestion process for the baked and steamed buns, respectively. The mathematical model presented an R^2^ over 0.98 for all cases, proving the excellent fit between the formula and the experimental data. The amount of FFA release during the digestion of buns made with olive or sunflower oleogels stabilized by HPMC and XG was around 35–48%. This result agrees with results obtained by other authors [40,41] who found that the amount of FFA released during the digestion of emulsions stabilized by HPMC was 44–51%. The HPMC interfacial activity and the lower destabilization under the digestion fluids plays a key role on lipolysis, decreasing the access of lipase to the interface [42]. Neither baked nor steamed buns made with oleogels (BO, BS, SO, and SS), showed significant (*p <* 0.05) differences in FFA_max_ values compared with their respective buns made with margarine (BC and SC) (Table 3 and Table 4).

Buns made with oleogels showed higher FFA release profiles than margarine along digestion (Figure 2) because of two factors: (I) margarine is formulated with emulsifiers, which inhibit the lipase activity more effectively than hydrocolloids; and (II) the presence of calcium ions from the digestion and the calcium presented in the food (from the milk used in the bun’s making) increases the removal of long-chain fatty acids from the oil–water interface by forming insoluble calcium soaps, thus allowing lipase access to the interface [43,44].

The initial digestion rate of the baked buns showed significant differences between BC and BS (Table 3) probably influenced by substituting saturated margarine fat, stabilized with surfactants, for sunflower oleogel rich in monounsaturated fatty acids and stabilized with hydrocolloids. The baked buns formulated with sunflower oleogels exhibited lower initial digestion rates, indicating than control buns were initially digested faster (Table 3). Commercial margarine was formulated with E-471 as a surfactant, a mixture of monoglycerides and diglycerides that are likely to contribute substrates for the action of intestinal lipase and generate additional FFAs and even serve as emulsifiers that boost the efficiency of lipolysis, increasing the initial rate [45]. No significant (*p >* 0.05) differences were found between BO and the rest of the samples (Table 3). For steamed buns, the initial rate was higher (*p <* 0.05) in the SS buns than in SC buns (Table 4). No significant (*p >* 0.05) differences were found between SO and the rest of the samples (Table 4).

The degree of digestion of lipids was not affected by the replacement of margarine by oleogels. However, the initial rate of lipid digestion was different depending on the type of processing used in bun making. These results have important implications for the rational design of foods and could be further investigated as a strategy to modulate lipid digestion for healthier foods.

## 4. Conclusions

The replacement of margarine, a plastic fat traditionally used in manufactured baked products, with olive or sunflower oil oleogels structured with HPMC and XG, was effective in providing similar physical characteristics in baked or steamed buns. Replacement of margarine for oleogels successfully produced steamed buns without differences in the crumb structure, volume, height, and texture. However, the baked buns made with oleogels had a less aerated crumb structure and were harder than the baked buns formulated with margarine. At the sensory level, no differences in texture were observed between the margarine buns and buns made with oleogels, both for baking and steaming conditions. Regarding lipid digestibility, the extent of digestion, as measured from the in vitro FFA release, was not affected by margarine replacement by oleogels. These results suggest that the reformulation of buns with oleogels as saturated fat replacers could help the food industry prepare healthy bakery products.

## Figures and Tables

**Figure 1 foods-10-01781-f001:**
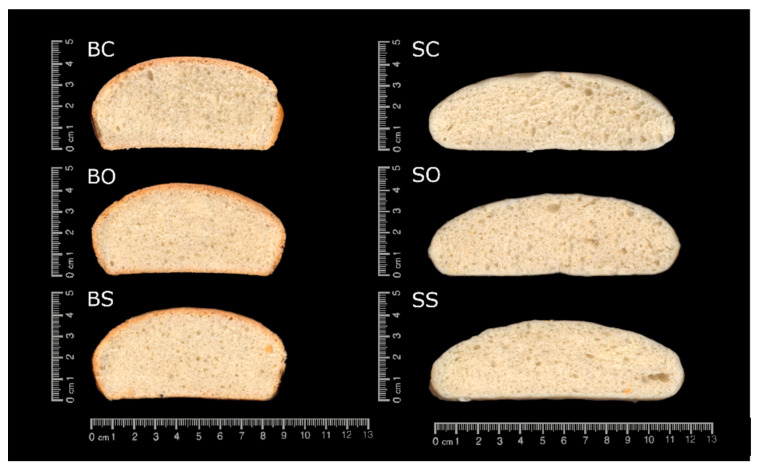
Cross section of breads. Baked control bun made with margarine (BC); baked bun made with olive oleogel (BO); baked bun made with sunflower oleogel (BS); steamed control bun made with margarine (SC); steamed bun made with olive oleogel (SO); steamed bun made with sunflower oleogel (SS).

**Figure 2 foods-10-01781-f002:**
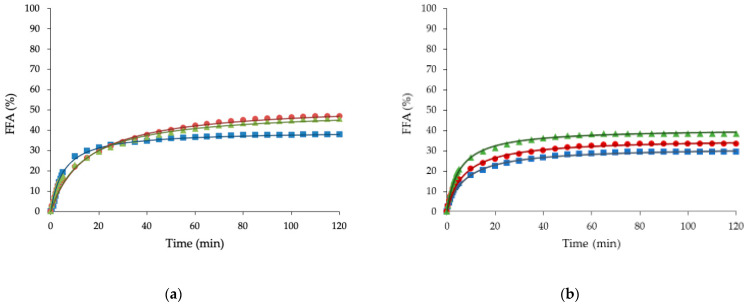
Free fatty acids (FFA) released during in vitro digestion of baked buns (**a**) and steamed buns (**b**). Control buns (squares), buns made with olive oleogel (circles), and buns made with sunflower oleogel (triangles).

**Table 1 foods-10-01781-t001:** Physical properties of baked buns.

Parameters		BC	BO	BS
Crumb Cell Structure	Cell density Cell area (mm^2^) Cell circularity Total cell area (%)	1235 ^a^ (113)	1045 ^a^ (140)	1064 ^a^ (100)
		0.27 ^a^ (0.06) 0.85 ^b^ (0.02) 45 ^b^ (3)	0.33 ^a^ (0.06) 0.83 ^a,b^ (0.01) 39.2 ^b^ (0.4)	0.30 ^a^ (0.01) 0.82 ^a^ (0.01) 36.6 ^a^ (0.7)
	Specific volume (cm^3^/g) Height (cm)	2.7 ^a^ (0.1)	2.54 ^a^ (0.09)	2.7 ^a^ (0.02)
		4.75 ^a^ (0.03)	4.7 ^a^ (0.1)	4.70 ^a^ (0.06)
Texture	Hardness (*n*) Springiness Cohesiveness Chewiness (*n*)	0.9 ^a^ (0.1)	1.2 ^b^ (0.2)	1.1 ^b^ (0.3)
		0.92 ^b^ (0.02)	0.90 ^a,b^ (0.03)	0.8 ^a^ (0.2)
		0.74 ^a^ (0.02)	0.73 ^a^ (0.02)	0.74 ^a^ (0.06)
		0.64 ^a^ (0.09)	0.8 ^b^ (0.1)	0.7 ^a,b^ (0.2)

Control baked bun made with margarine (BC); baked bun made with olive oleogel (BO); baked bun made with sunflower oleogel (BS). Values with different lowercase letters (^a, b^) within the same row are significantly different (*p <* 0.05) according to the LSD multiple range test.

**Table 2 foods-10-01781-t002:** Physical properties of steamed buns.

Parameters		SC	SO	SS
Crumb Cell Structure	Cell density Cell area (mm^2^) Cell circularity Total cell area (%)	1329 ^a^ (87)	1101 ^a^ (161)	1137 ^a^ (105)
		0.27 ^a^ (0.06) 0.85 ^a^ (0.01) 44 ^a^ (4)	0.27 ^a^ (0.06) 0.82 ^a^ (0.02) 37 ^a^ (6)	0.30 ^a^ (0.01) 0.84 ^a^ (0.01) 43 ^a^ (2)
	Specific volume (cm^3^/g) Height (cm)	2.50 ^a^ (0.05)	2.7 ^a^ (0.1)	2.6 ^a^ (0.2)
		3.9 ^a^ (0.3)	4.0 ^a^ (0.1)	3.8 ^a^ (0.01)
Texture	Hardness (*n*) Springiness Cohesiveness Chewiness (*n*)	0.7 ^a^ (0.1)	0.75 ^a^ (0.08)	0.7 ^a^ (0.1)
		0.90 ^a^ (0.03)	0.91 ^a^ (0.03)	0.9 ^a^ (0.1)
		0.72 ^a^ (0.02)	0.72 ^a^ (0.03)	0.74 ^a^ (0.05)
		0.48 ^a^ (0.08)	0.49 ^a^ (0.05)	0.5 ^a^ (0.1)

Control steamed bun made with margarine (SC); steamed bun made with olive oleogel (SO); steamed bun made with sunflower oleogel (SS). Values with lowercase letter (^a^) within the same row are significantly different (*p <* 0.05) according to the LSD multiple range test.

**Table 3 foods-10-01781-t003:** Kinetic parameters of lipid digestion in baked buns.

	FFA_max_ (%)	K (1/min)	R^2^
BC	38 ^a^ (2)	6.5 ^b^ (1.3)	0.99
BO	48 ^a^ (3)	4.2 ^a,b^ (0.7)	0.99
BS	45 ^a^ (7)	3.7 ^a^ (0.1)	0.98

FFA_max_ (%): percentage of maximum free fatty acids released; K: initial rate of free fatty acid release. Control baked bun made with margarine fat (BC); baked bun made with olive oleogel (BO); baked bun made with sunflower oleogel (BS). Values with different lowercase letters (^a, b^) within the same column are significantly different (*p <* 0.05) according to the LSD multiple range test.

**Table 4 foods-10-01781-t004:** Kinetic parameters of lipid digestion in steamed buns.

	FFA_max_ (%)	K (1/min)	R^2^
SC	31 ^a^ (8)	3.8 ^b^ (0.6)	0.99
SO	35.7 ^a^ (0.4)	5.5 ^a,b^ (0.6)	0.99
SS	39 ^a^ (4)	6.8 ^b^ (1.4)	0.99

FFA_max_ (%): percentage of maximum free fatty acids released; K: initial rate of free fatty acid release. Control steamed bun made with margarine fat (SC); steamed bun made with olive oleogel (SO); steamed bun made with sunflower oleogel (SS). Values with different lowercase letters (^a, b^) within the same column are significantly different (*p <* 0.05) according to the LSD multiple range test.

## Data Availability

Research data are not shared.

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
