# Peer review of "Use of Oleogels to Replace Margarine in Steamed and Baked Buns"

_foods, 2021, doi:10.3390/foods10081781_

Round 1
Reviewer 1 Report
This article analyzes the use of oleogels, made with sunflower and olive oil, as a replacer for margarine in baked and steamed buns. The article is very well written and provides a good insight on the potential oleogels have in producing healthier food products. What I found particularly nice is that all equipments used to produce the oleogels and the buns are commonly found in households, therefore, this study shows the potential of the use of oleogels not only for food industries but also for consumers to produce at home. Therefore I would suggest that author do mention this extra positive point in their article. In the introduction I would suggest author add one more sentence describing what is an oleogel.
Other minor suggestions:
Line 45: replacement
Line 132 and 137:panelists in a better word for that
Section 3.1: Use past tense when discussing results, change "have" to "had" throughout the discussion
Table 1: As in the methodology author write the TAP analysis was carried out in triplicate I suggest that they add the standard errors here so that readers have a better idea of the differences between the samples
Author Response
This article analyzes the use of oleogels, made with sunflower and olive oil, as a replacer for margarine in baked and steamed buns. The article is very well written and provides a good insight on the potential oleogels have in producing healthier food products. What I found particularly nice is that all equipments used to produce the oleogels and the buns are commonly found in households, therefore, this study shows the potential of the use of oleogels not only for food industries but also for consumers to produce at home. Therefore I would suggest that author do mention this extra positive point in their article. In the introduction I would suggest author add one more sentence describing what is an oleogel.
Thank you for your comments. A sentence related to the potential of using oleogels by consumers has been added (lines 67-68) the explanation describing what is an oleogel has been added as well (lines 48-50).
Other minor suggestions:
Line 45: replacement
According to the reviewer suggestion, the word ‘replace’ has been modified in the line 45.
Line 132 and 137: panelists in a better word for that
The word “taster” has been changed in the lines 135 and 140 to the word “panelists/panelist”.
Section 3.1: Use past tense when discussing results, change "have" to "had" throughout the discussion
The discussion of the results has been rewritten using past tense (lines 188-189).
Table 1: As in the methodology author write the TAP analysis was carried out in triplicate I suggest that they add the standard errors here so that readers have a better idea of the differences between the samples
Standard deviations are included in the table 1
Reviewer 2 Report
Review comments for manuscript
“Use of oleogels to replace margarine in steamed and baked buns»
The manuscript “Use of oleogels to replace margarine in steamed and baked 2 buns” technology” by Bascuas et al. is a very interesting, well written paper.
Below are some comments that may help authors to improve their manuscript:
Line 169: The reference for Equation 1 is [26]?
Line 171: “Mlipid is the average molecular weight of solid fat” What was that? The value has to be reported.
3.4. SENSORY ANALYSIS SECTION: If there are any comments regarding the acceptance/likeness of the bans should be mentioned.
Hedonic test sensory evaluation should be helpful to understand if the panelists liked and were keen to consume the new products
Lines 286-291: Some information on lipase activity should be added in order the behavior of FFA be better explained.
Figure 2: I suggest the authors to keep the same colors for the different samples between the baked and steamed buns, e.g. blue for control, green for those prepared with olive oil oleogel etc.
Lines 312-323: What's the importance/practical meaning of these results? Please, explain. Also comment on the different behavior of baked and steamed buns.
Author Response
The manuscript “Use of oleogels to replace margarine in steamed and baked 2 buns” technology” by Bascuas et al. is a very interesting, well written paper.
Below are some comments that may help authors to improve their manuscript:
Line 169: The reference for Equation 1 is [26]?
The reference for Equation 1 is [27]. It has been added in line 171.
Line 171: “Mlipid is the average molecular weight of solid fat” What was that? The value has to be reported.
The values were 200g/mol for margarine and 282 g/mol for olive and sunflower oil oleogels According to the reviewer suggestion, the values have been added in line 175-176.
3.4. SENSORY ANALYSIS SECTION: If there are any comments regarding the acceptance/likeness of the bans should be mentioned. Hedonic test sensory evaluation should be helpful to understand if the panelists liked and were keen to consume the new products
We agree with the reviewer on the usefulness of the hedonic test. In fact, during the experimental design, we selected the hedonic sensory test to be performed in our bun samples. However, due to the pandemic situation, we were unable to perform this type of sensory evaluation that requires a high number of consumers, which were not available.
Lines 286-291: Some information on lipase activity should be added in order the behavior of FFA be better explained.
According with the reviewer, the information has been added in lines 290-292.
Figure 2: I suggest the authors to keep the same colors for the different samples between the baked and steamed buns, e.g. blue for control, green for those prepared with olive oil oleogel etc.
According to the reviewer’s suggestion, the colors have been changed.
Lines 312-323: What's the importance/practical meaning of these results? Please, explain. Also comment on the different behavior of baked and steamed buns.
Comments on the importance and practical meaning of the results have been included in lines 328-332. However, we prefer not including comments on the different behavior of baked and steamed buns as comparing the processes is not included in the objectives of the work.
Round 2
Reviewer 2 Report
The manuscript can be now published in Foods journal.